# Photodynamic Therapy of Aluminum Phthalocyanine Tetra Sodium 2-Mercaptoacetate Linked to PEGylated Copper–Gold Bimetallic Nanoparticles on Colon Cancer Cells

**DOI:** 10.3390/ijms24031902

**Published:** 2023-01-18

**Authors:** Nokuphila Winifred Nompumelelo Simelane, Gauta Gold Matlou, Heidi Abrahamse

**Affiliations:** Laser Research Centre, Faculty of Health Sciences, University of Johannesburg, P.O. Box 17011, Johannesburg 2028, South Africa

**Keywords:** aluminum phthalocyanine, photodynamic therapy, PEGylated nanoparticles, colon cancer

## Abstract

This work reports for the first time on the synthesis, characterization, and photodynamic therapy efficacy of the novel aluminium (III) chloride 2(3), 9(10), 16(17), 23(24)-tetrakis-(sodium 2-mercaptoacetate) phthalocyanine (AlClPcTS41) when alone and when conjugated to PEGylated copper–gold bimetallic nanoparticles (PEG-CuAuNPs) as photosensitizers on colon cancer cells (Caco-2). The novel AlClPcTS41 was covalently linked to the PEG-CuAuNPs via an amide bond to form AlClPcTS41-PEG-CuAuNPs. The amide bond was successfully confirmed using FTIR while the crystal structures were studied using XRD. The morphological and size variations of the PEG-CuAuNPs and AlClPcTS41-PEG-CuAuNPs were studied using TEM, while the hydrodynamic sizes and polydispersity of the particles were confirmed using DLS. The ground state electron absorption spectra were also studied and confirmed the typical absorption of metallated phthalocyanines and their nanoparticle conjugates. Subsequently, the subcellular uptake, cellular proliferation, and PDT anti-tumor effect of AlClPcTS41, PEG-CuAuNPs, and AlClPcTS41-PEG-CuAuNPs were investigated within in vitro Caco-2 cells. The designed AlClPcTS41 and AlClPcTS41-PEG-CuAuNPs demonstrated significant ROS generation abilities that led to the PDT effect with a significantly decreased viable cell population after PDT treatment. These results demonstrate that the novel AlClPcTS41 and AlClPcTS41-PEG-CuAuNPs had remarkable PDT effects against Caco-2 cells and may trigger apoptosis cell death pathway, indicating the potential of the AlClPcTS41 and AlClPcTS41-PEG-CuAuNPs in enhancing the cytotoxic effect of PDT treatment.

## 1. Introduction

Colorectal cancer (CRC) has become one of the utmost challenging malignancies and is ranked as the second highest lethal cancer among the global population [1]. In the past decade, CRC incidence rate has been on the rise, and in just the year 2020, CRC incidence rate was estimated to have accounted for approximately 10% of global cancer incidence [1]. Several common CRC treatment modalities include invasive surgery, chemotherapy, and radiotherapy [2]. Nevertheless, these therapeutic modalities still lack a complete cure for CRC due to their shortcomings such as invasiveness, undesirable systemic side effects, and toxicity [2]. Accordingly, there is an unmet need to obtain more improved and effective therapeutic outcomes for CRC, and potentially reduce side effects; thus, an urgent need for the development of prospective CRC treatment is required [2]. Photodynamic therapy (PDT) is a developing antitumor approach that is less harmful to patients, selectively eradicates cancerous cells with minimal side effects, and is tolerable to repeated doses with far more effectiveness as compared to the conventional CRC treatment [3]. 

PDT is a novel cancer treatment method that applies the chemical reaction of a photosensitizer (PS) agent, activated with light irradiation of a particular wavelength, within an environment that contains molecular oxygen to yield cytotoxic oxygen species responsible for the induction of cellular death [4]. Metalized phthalocyanines (MPcs) are used extensively as photosensitizer agents in PDT due to their ability to absorb irradiation light and undergo photochemical and photophysical pathways to generate high yields of cytotoxic singlet oxygen and other reactive oxygen species [5,6,7]. Interestingly, the central metalation within the MPcs cavity is reported to populate excited molecules through to the excited triplet state through intersystem crossing by heavy atom effect that also improves the yield of cytotoxic singlet oxygen [8,9,10]; hence, aluminum is used as a central metal in this study. Derivatives of MPcs are typically achieved by the addition of different substituents on the fourth (peripheral (β)) and third (non-peripheral (α)) positions of the isoindoline subunits (benzene rings), which can improve the properties of the MPcs [11,12,13]. In this work, sodium 2-mercaptoacetate is used as a target substituent on the fourth (peripheral) position of the benzene rings of the macrocyclic MPc structure. The salt form improves the water solubility of the complex and makes it easy to apply in PDT and subcellular localization studies [14]. 

Among the well-studied MPcs, aluminum phthalocyanines (AlClPc) have been applied as PS agents for various solid cancerous cancer such as breast, colon, and oesophageal [15]. AlClPc presents a strong absorption of visible light at the wavelength between 650–800 nm where tissue penetration is maximized, which is crucial for PDT since it can allow for efficient treatment of cancerous tissues with negligible phototoxicity. In vitro studies using aluminum (III) phthalocyanine chloride tetra sulphonate (AlPcS4) photosensitizers have been previously reported [15,16]. Chizenga et al. (2019) demonstrated a notable dose-dependent decrease in cell proliferation and increased cytotoxicity of cervical cancer cells and cervical CSCs post-irradiation, with a 673.2 nm diode laser [15]. It has also been reported that AlPc-induced PDT can significantly enhance the singlet oxygen quantum yields and efficiently damage cell membranes on proteins [16]; hence, AlClPc derivative is used in this study. Nonetheless, a majority of the existing MPcs are insoluble in aqueous media and suffer from extensive aggregation, which could hamper their phototoxicity in application [4]. The thiol group (sulfur) of the MPcs can further improve their water solubility and prevent them from aggregating, without significantly hampering their photophysical properties [17].

In recent years, the development of effective delivery approaches, such as nanoparticles, as carriers for PS delivery has become indispensable [18]. In this regard, the anti tumor efficacy of conventional MPcs, such as aluminum phthalocyanines (AlClPc), can be improved upon by incorporating them in nanocarriers, particularly bimetallic nanoparticles, such as copper–gold nanoparticles, to enhance PS drug delivery [15,19,20]. Among the metallic nanoparticles, one of the increasingly favorable PS carriers within PDT-induced colorectal cancer treatment are gold nanoparticles (AuNPs), principally because they possess unique properties, such as distinctive miniature dimensions and a large surface area-to-volume ratio [21]. These nanoparticles (NPs) in different forms may passively localize in cancerous cells owing to certain tumor characteristics, such as leaky vasculature and poor lymphatic drainage, a phenomenon generally known as the enhanced permeability and retention effect (EPR) [18,21,22]. Studies by Caro et al. (2021) have also reported that the EPR effect is also highly dependent on the type of tumor and tumor site as some factors to consider for an efficient tumor targeting strategy [23,24]. Furthermore, to address the serious biological barriers, such as macrophages, often encountered by NPs, polyethylene glycol (PEG) is used as a surface modification polymer to improve the solubility of nanoparticles, reduce their nonspecific interactions with the biological barriers, improve accumulation target disease, reduce cell binding, uptake, as well as interfere with aggregation, thus enhancing surface hydrophilicity [25,26]. PEG molecules are also used to facilitate covalent linkage between the novel MPc derivative and the copper–gold bimetallic nanoparticles. 

Therefore, this study aims to investigate the PDT effects of a novel photosensitizer (AlClPcTS41) when alone and when conjugated to PEGylated copper–gold bimetallic (alloyed) nanoparticles (PEG-CuAuNPs) (AlClPcTS41-PEG-CuAuNPs) by investigating morphological changes, dose–response studies, and ATP cell proliferation assays on human colon cancer (Caco-2) cell line and cell death mechanisms using flow cytometry.

## 2. Results

### 2.1. Synthesis of AlClPcTS41

The synthesis of AlClPcTS41 was achieved by reacting the phthalonitrile precursor in the presence of a base (DBU) and aluminum (III) chloride to afford the phthalocyanine complex, as previously reported [12,27,28,29], as seen in Scheme in Section 4.2.1. FTIR showed the disappearance of the nitrile (-CN) peak on the infrared spectra of AlClPcTS41, confirming the successful cyclotetramerization of the phthalonitrile precursor into the phthalocyanine complex during the reaction [9,29,30], as seen in Figure 1. The FTIR spectra also showed a stretching peak at 1716 cm^−1^ (C=O), a broad stretching peak at 3315 cm^−1^ (-OH), stretching peaks at 1560 cm^−1^ (Ar-C=C), 1610 cm^−1^ (C=C), 2735 cm^−1^ (Ar-S-C), and 2950 cm^−1^ (Ar-CH) corresponding and proving various functional groups of the AlClPcTS41 complex, as seen in Figure 1. The 1H NMR spectra were obtained in deuterated water (D_2_O), as seen in Appendix A. The cyclic structure of the AlClPcTS41 consisting of aromatic rings gave peaks between 7.63 ppm and 8.13 ppm, while the hydroxyl group of the acetate substituent was identified by a singlet peak at 8.44 ppm integrating to four protons. The other functional groups of the sodium 2-mercaptoacetate were identified by a singlet peak at 3.59 ppm that integrated into eight protons, making up the structure of AlClPcTS41. MALDI-TOF mass spectrum of the AlClPcTS41 gave the mass closer to the calculated mass unit. Elemental analysis of AlClPcTS41 gave values in agreement with analytically calculated values. 

### 2.2. Characterization of Nanoparticles and Conjugates 

#### 2.2.1. FTIR

PEG-CuAuNPs were synthesized by a chemical reduction by first synthesizing the copper nanoparticles followed by coating them with gold while stabilizing them with sodium citrate. PEGylation of the citrate-stabilized copper–gold alloy nanoparticles was achieved by ligand exchange between the citrate molecules and PEG molecules. FTIR showed an intense stretching peak at ca. ~2900 cm^−1^, typical of saturated carbons that are present in the PEG polymer of the PEG-CuAuNPs, as well as the amino groups stretching peaks at 3400 cm^−1^, as seen in Figure 1. The amide bond (covalent) linkage of the amino groups of the PEG-CuAuNPs and the carboxylate groups of the AlClPcTS41 were successfully demonstrated by the NH bend of an amide bond at 1610 cm^−1^, an amide carbonyl carbon peak at 1690 cm^−1^, and a secondary NH stretch at 3356 cm^−1^, as seen in Figure 1. These confirmed the chemical linkage of the PEG-CuAuNPs and AlClPcTS41 to form the covalently linked AlClPcTS41-PEG-CuAuNPs.

**Figure 1 ijms-24-01902-f001:**
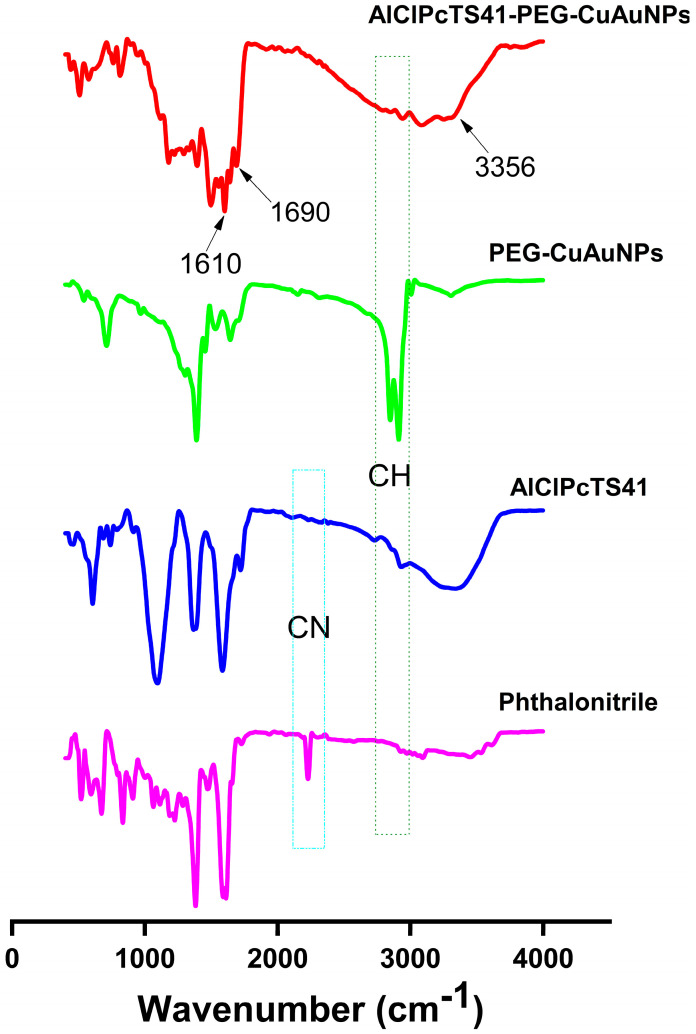
Fourier-transform infrared spectroscopy of the chemically linked AlClPcTS41-PEG-CuAuNPs compared to the AlClPcTS41 and PEG-CuAuNPs alone.

#### 2.2.2. TEM 

TEM micrographs were used to study the size and morphology changes of the nanoparticle (PEG-CuAuNPs) and conjugates (AlClPcTS41-PEG-CuAuNPs), as seen in Figure 2. PEGylated bimetallic (alloy) nanoparticles with spherically shaped sizes between 10 nm and 20 nm (average 13.24 nm) were successfully synthesized using a method previously described. The PEG molecules were linked through the Au-S affinity on the surface of the intermetallic gold of the alloy nanoparticles. The PEG-CuAuNPs were then linked to the AlClPcTS41 through an amide bond between the amino groups of the PEG molecules of the nanoparticles surface and the substituents carboxylate salt groups of the phthalocyanine complexes, as confirmed using FTIR, as seen in Figure 1 and Scheme in Section 4.2.3. The size of the conjugates increased (15 nm–35 nm range) (average 19.36 nm) as compared to the nanoparticle alone (10 nm–20 nm) range, as seen in Figure 2. The addition of the phthalocyanine complexes on the nanoparticle surface resulted in amorphous structures symbolic of the carbon backbone of the phthalocyanine complex (AlClPcTS41), as seen in Figure 2B. Phthalocyanine molecules form co-planar associations with other rings of phthalocyanines on the surface of nanoparticles, as well as on adjacent nanoparticles, forming aggregates that result in an increase in size for the conjugates (AlClPcTS41-PEG-CuAuNPs) as compared to the nanoparticles alone (PEG-CuAuNPs) [31,32,33].

#### 2.2.3. XRD 

The crystal structure of the nanoparticle and the conjugates were investigated using X-ray diffraction spectroscopy, as seen in Figure 3. The PEG-CuAuNPs showed crystal peaks at 2θ = 38.5° (Au), 44.8° (Cu), 65.0° (Au), 74.0° (Cu), 77.8° (Au), and 81.9° (Au), corresponding to 111 (Au), 111 (Cu), 220 (Au), 220 (Cu), 311 (Au), and 222 (Au) crystal planes of copper–gold bimetallic nanoparticles, as seen in Figure 3. AlClPcTS41 alone showed a broad peak at 2θ = 15°, which is typical of amorphous carbon of the phthalocyanine ring [34]. AlClPcTS41-PEG-CuAuNPs showed both crystal peaks of PEG-CuAuNPs and the broad peak of the phthalocyanine, as seen in Figure 3. The size diameter of the PEG-CuAuNPs and the AlClPcTS41-PEG-CuAuNPs were determined using the Debye-Scherrer Equation (1) [35]: (1)$$d=\frac{k\lambda }{\beta Cos\theta }$$
where d is the crystal size, λ is the wavelength of the X-ray source (0.1541 nm), k is an empirical constant equal to 0.9, β is the full width at half maximum of the diffraction peak, and θ is the diffraction angle of the crystal orientation peak. There was an increase in XRD sizes of PEG-CuAuNPs (12.76 nm) to the AlClPcTS41-PEG-CuAuNPs (18.12 nm), Table 1. An increase in the size of the alloy nanoparticles compared to the phthalocyanine-nanoparticle conjugates is due to aggregates of phthalocyanines that form through π–π stacking [31] on the surface of nanoparticles, Table 1.

#### 2.2.4. DLS

Dynamic light scattering studies the hydrodynamic size distributions of particles in the solution. The hydrodynamic sizes of the PEG-CuAuNPs and AlClPcTS41-PEG-CuAuNPs were found to be between 9 nm and 52 nm (average 18.71 nm) and between 15 nm and 42 nm (average 24.83 nm), respectively, Figure 4. A size increase in the conjugates (AlClPcTS41-PEG-CuAuNPs), as compared to PEG-CuAuNPs, is due to the incorporation of phthalocyanine complexes on the surface of nanoparticles. Phthalocyanines are known to form aggregates through π–π stacking [31] with each other which results in an increase in size when conjugated to nanoparticles. The polydispersity index (PDI) of the PEG-CuAuNPs (0.214) and the AlClPcTS41-PEG-CuAuNPs (0.258) are listed in Table 1. Both PDI values are an indication of good size distribution and below the 0.7 PDI indicates lower particle size distribution [36]. DLS is known to be inclined to bigger sizes as compared to other techniques [37]; thus, the sizes reported for TEM and XRD are smaller than those reported from DLS, Table 1.

#### 2.2.5. UV–VIS 

UV–VIS was used to study the electronic absorption spectra of the novel AlClPcTS41, which were investigated in water and DMSO, while the PEG-CuAuNPs and AlClPcTS41-PEG-CuAuNPs absorption spectrums were obtained in water, as seen in Figure 5. The absorption spectra of AlClPcTS41 were typical of electronic absorption spectra of metallated phthalocyanine complexes in DMSO [38,39]. The Q-band maxima of the AlClPcTS41 was 715 nm in DMSO and 642 nm in water, with the B-band maxima at 365 nm in DMSO and 360 nm in water, as seen in Figure 5. The high refractive index of the DMSO as compared to the water is known to result in red-shifting of the Q-band, as in the case of AlClPcTS41 [40]. The ability of the phthalocyanine derivative (AlClPcTS41) to absorb light within the therapeutic window (600–900 nm) is an important characteristic for a photosensitizer to undergo photochemical reactions after excitation that will yield cytotoxic singlet oxygen responsible for cancer cell death [41,42]. PEG-CuAuNPs electronic spectra showed a weak surface plasmon resonance (SPR) peak at ca. 480–550 nm for the copper–gold nanoparticles. Upon conjugation of the AlClPcTS41 to the PEGylated copper–gold bimetallic nanoparticles, there was a broadening of the absorption spectra for the AlClPcTS41-PEG-CuAuNPs conjugates below 550 nm due to the SPR band of the nanoparticles, as seen in Figure 5. The Q-band of the AlClPcTS41-PEG-CuAuNPs remained the same as compared to the Q-band of AlClPcTS41 in water, as seen in Figure 5. 

### 2.3. Photodynamic Therapy Dose Response

#### 2.3.1. Morphology

Typical images of Caco-2 cells obtained before and after irradiation of the different treatments, as well as control groups, are demonstrated in Appendix A. Our control experiments (dark toxicity) of Caco-2 cells that were not subjected to laser irradiation with an excitation laser diode at 636 nm demonstrated insignificant changes in the morphology of cells. However, in cells that received doses (0.125, 0.25, 0.5, and 0.75 µM) of AlClPcTS41 and lasers, an increase in doses resulted in observable changes in morphology. The morphology of the treated cells showed signs of floating in the medium with a clear indication of agglomerations to each other. This was further supported by the substantial decrease in the number of living cells within the treated groups, as seen in Appendix A.

#### 2.3.2. Viability

The relative cell viability of Caco-2 cells incubated with various concentrations (0.125, 0.25, 0.5, and 0.75 µM) of the AlClPcTS41 with or without laser irradiation are illustrated in Figure 6. Caco-2 cell survival was not affected by the absence of light or AlClPcTS41 alone. However, a trypan blue dye exclusion assay showed that the photosensitizer followed by irradiation (10 J/cm^2^) induced a statistically significant reduction in cell viability. Within the treated groups, a significant decrease in cell survival, approximately between 80–40% in a dose–response manner, was observed as opposed to the untreated control on all the AlClPcTS41 concentrations administered, indicating that AlClPcTS41-PDT was more phototoxic to Caco-2 cells during treatment, as seen in Figure 6. Almost 54% viability was observed at a 0.5 µM concentration, after exposure to laser irradiation. Based on these findings, the lowest inhibitory concentration (IC50) was then established by calculations, using a simple linear dose–response model from data extracted from the trypan blue viability biochemical assay conducted, as seen in Figure 7. The optimal IC50 concentration of AlClPcTS41 during PDT treatment was found to be 0,58 µM in our group, which was subsequently used in further experiments. Within PDT treatment, the efficacy of the AlClPcTS41-PEG-CuAuNPs was compared to cells only and quantified by the trypan blue exclusion assay, as seen in Figure 8. The results showed that, in the absence of laser light irradiation, insignificant cell toxicity was observed within the control groups, suggesting that free AlClPcTS41, PEG-CuAuNPs, and/or AlClPcTS41-PEG-CuAuNPs are non-cytotoxic for the experimental conditions utilized. However, upon laser light irradiation at 636 nm, the Caco-2 cells treated with AlClPcTS41 and AlClPcTS41-PEG-CuAuNPs had 48.7% and 35.8% of viable cells, respectively, suggesting evidence of improved cytotoxic effects.

#### 2.3.3. Cytotoxicity

The result of the LDH assay showed 20%, 45%, 55%, and 85% significant increases, in a dosed manner, in the levels of LDH within dosages of 0.125, 0.25, 0.5, and 0.75 µM of the AlClPcTS41, respectively, after treatment with the photosensitizer and exposure to laser light as compared to control groups, as seen in Figure 9. Additionally, LDH release was decreased in cells only in the control group, by 6%, while between 5–9% decreased levels of LDH were also achieved in either photo-inactivated control groups, non-irradiated groups, AlClPcTS41 only or irradiation-only control groups. Treatment groups for Caco-2 cells, as seen in Figure 9, with doses of 0.125, 0.25, 0.5 and 0.75 µM showed statistically significant results of *p* < 0.001 (***).

### 2.4. Cellular Localization of AlClPcTS41 and AlClPcTS41-PEG-CuAuNPs Nanoconjugates 

The intracellular localization of AlClPcTS41 and AlClPcTS41-PEG-CuAuNPs was studied by using fluorescence microscopy. PEG-CuAuNPs demonstrated no phototoxicity; hence, its intracellular localization was not studied. Figure 10 shows the blue fluorescence emanated from DAPI, which stained the cell nucleus, and two intracellular organelles (lysosome and mitochondria) were stained with organelle-specific fluorescent tracker dyes. Figure 10 also shows that the MitoTracker and LysoTracker dyes were stained green, and the AlClPcTS41 was marked in red fluorescence, to indicate the PS localization in different intracellular organelles. The localization of AlClPcTS41 (red fluorescence) in lysosome and mitochondria displayed a yellow fluorescence in the merged images, as illustrated in Figure 10. Comparisons between the AlClPcTS41 distribution evinced that the ALClPcTS41 most likely localized in both the mitochondria and lysosomes and it turned out that the AlClPcTS41-PEG-CuAuNPs appeared to have accumulated more in both the intracellular organelles.

### 2.5. AlClPcTS41 and AlClPcTS41-PEG-CuAuNPs-Mediated PDT 

#### 2.5.1. Morphology

The morphological changes in Caco-2cells after AlClPcTS41, PEG-CuAuNPs, and AlClPcTS41-PEG-CuAuNPs-mediated PDT were examined under an inverted microscope, as seen in Figure 11. Images illustrate great disruption of Caco-2 morphology in AlClPcTS41 and AlClPcTS41-PEG-CuAuNPs-PDT treated experimental groups, as seen in Figure 11, as compared to PEG-CuAuNPs and control groups. Moreover, Caco-2 cells exposed to AlClPcTS41 and AlClPcTS41-PEG-CuAuNPs-PDT exhibited significant morphological changes such as cell shrinkage, cell rounding up, loss of membrane integrity, vacuole formation, and cell density decrease and cell detachment from each other, when compared to PEG-CuAuNPs and control groups. In comparison, the untreated control cells and laser irradiation alone control group were found to be similar across the control group; they displayed no morphological changes. Caco2 cells were equally distributed and grew.

#### 2.5.2. Cytotoxicity

The release of lactate dehydrogenase (LDH) into media for estimating damage to the cell membrane after AlClPcTS41, PEG-CuAuNPs, and AlClPcTS41-PEG-CuAuNPs-based photodynamic treatment was investigated, as seen in Figure 12. In our study, the cytotoxic effects of AlClPcTS41, PEG-CuAuNPs, and AlClPcTS41-PEG-CuAuNPs on Cac0-2 cells were determined after excitation with light at a wavelength of 636 nm, and an LDH assay was used. Firstly, before determining the PDT efficacy of AlClPcTS41, PEG-CuAuNPs, and AlClPcTS41-PEG-CuAuNPs, the dark cytotoxicity of the drugs was investigated. Our LDH assay showed that the dark cytotoxicity control groups consisting of free AlClPcTS41 and AlClPcTS41-PEG-CuAuNPs exhibited high levels of cell viability, as observed in Figure 12, indicating negligible toxicity. However, under 636 nm laser irradiation, fluency 10 J/cm^2^, it was found that the cytotoxicity level increased for AlClPcTS41 to 45%, as seen in Figure 12. Following treatment with AlClPcTS41-PEG-CuAuNPs, cytotoxicity rates of Caco-2 cells exposed to photodynamic irradiation wavelength of 636 nm increased to approximately 65%. When the concentration of AlClPcTS41 is 0.58 µM, statistically significant increases in LDH levels were detected in AlClPcTS41-PEG-CuAuNPs in Caco-2 cells compared to PEG-CuAuNPs or free AlClPcTS41. Under these laser irradiation conditions, the capabilities of AlClPcTS41-PEG-CuAuNPs-mediated PDT were better than those of AlClPcTS41-mediated PDT only.

#### 2.5.3. Proliferation 

The effects of AlClPcTS41, PEG-CuAuNPs, and AlClPcTS41-PEG-CuAuNPs on cell proliferation were determined using an ATP assay, as seen in Figure 13. The cell proliferation of Caco-2 incubated in the presence of laser irradiation alone; AlClPcTS41, PEG-CuAuNPs, and AlClPcTS41-PEG-CuAuNPs alone did not decrease significantly and the cell viability was close to the value observed for Caco-2 cells incubated in the absence of laser light (dark toxicity). The results showed that neither irradiation light, free AlClPcTS41, PEG-CuAuNPs, or AlClPcTS41-PEG-CuAuNPs had significant killing effects on cell viability, as seen in Figure 13. The cell proliferation rate of Caco-2 cells incubated with AlClPcTS41 was only at 0.58 µM and was high enough (80% ± 0.3%) to consider the photosensitizer as nontoxic in the absence of light irradiation. In comparison with these results, a substantial reduction in cell proliferation was observed when Caco-2 cells were treated with AlClPcTS41 or AlClPcTS41-PEG-Cu-AuNPs and subsequently exposed to laser irradiation of 636 nm at a fluency of 10 J/cm^2^; the cell proliferation of Caco-2 were, respectively, 55% and 37% (*** *p* < 0.001), suggesting the unique killing effects of AlClPcTS41-PEG-CuAuNPs after nano photodynamic treatment. 

#### 2.5.4. Cell Death Mechanism Analysis 

To assess whether AlClPcTS41, PEG-CuAuNPs, and AlClPcTS41-PEG-CuAuNPs-mediated PDT treatment elicited enhanced apoptosis in Caco2 cells, cells suspension from the various treatment and control Caco-2 populations were stained with Annexin V (a marker for early apoptotic) and PI (a marker for late apoptotic), and subsequently analyzed using flow cytometry. Our findings showed that AlClPcTS41 induced significant apoptosis of Caco-2 cells as compared to PEG-CuAuNPs, as seen in Figure 14. AlClPcTS41-PEG-CuAuNPs treatment of 0.58 µM (ICD 50) plus a 10 J/cm^2^ light dose resulted in a further increased apoptosis in Caco-2 cells than AlClPcTS41-PDT alone; the apoptosis rates were (11.050 ± 0.3%). This increase was statistically significant compared with the apoptosis of Caco-2 cells induced by AlClPcTS41-PDT. It was found that AlClPcTS41-PEG-CuAuNPs-PDT significantly promoted apoptosis (early and late) of Caco-2 cells compared to the cell control group, which exhibited only 4.40% apoptosis in Caco-2 cells. In addition, the apoptosis rate was not statistically different in the AlClPcTS41 alone group nor the laser alone group of Caco-2 cells. Statistical analysis of the apoptosis rate among groups is illustrated in Figure 14.

#### 2.5.5. Determination of Intracellular ROS Levels 

To assess the PDT effects of AlClPcTS41 and/or AlClPcTS41-PEG-CuAuNPs nanoconjugates in the presence or absence of light irradiation, the level of intracellular ROS production was determined by ROS-Glo H_2_O_2_ analysis. As shown in Figure 15, incubating with AlClPcTS41, PEG-CuAuNPs and/or AlClPcTS41-PEG-CuAuNPs, without irradiation, did not increase the level of ROS production compared to AlClPcTS41 or the AlClPcTS41-PEG-CuAuNPs-treated experimental group. However, it was observed that compared to the cell-only control, the free AlClPcTS41 plus irradiation group demonstrated an increase in ROS intensity and these results were in correlation with flow cytometry analysis. We found that cells treated with AlClPcTS41-PEG-CuAuNPs plus irradiation rendered an even better effect, as they exhibited a significant increase in the level of intracellular ROS production when compared to the control group. These findings may be related to the presence of PEG-CuAuNPs, which most likely improved the passive delivery of the PS to the targeted cancerous tissues, and thus improved the overall PDT efficacy. 

## 3. Discussion

PDT has recently been demonstrated to be an effective therapeutic technique for malignant diseases such as skin cancer, colorectal cancer, and lung cancer [43]. While the overall classical PDT treatment modality has limitations, such as its hydrophobic nature and tendency to aggregate under a physiological environment of conventional PSs, which could result in the decreased therapeutic effectiveness of PDT [18], nanoparticle-mediated therapy is designed to overcome such obstacles [44,45]. The utilization of nanoparticles, such as copper–gold bimetallic nanoparticles, as nanocarrier delivery systems for PS is currently being actively investigated to improve the delivery of PS and minimize off-target PS delivery [46]. In addition, the use of polyethylene glycol (PEG) modified to the surface of NPs may also improve some desirable qualities, such as reducing their interference with biological barriers, thus improving the PS delivery to Caco-2 cells [18,44]. Here, we developed and synthesized a novel PS (AlClPcTS41)-conjugated onto polyethylene glycol (PEG) functionalized copper–gold nanoparticles (PEG-CuAuNPs) to form AlClPcTS41-PEG-CuAuNPs conjugates and studied their phototoxic effect on targeted PDT of in vitro cultured Caco-2 cells.

The subcellular localization of a photosensitizer has been shown to be paramount for the eradication of cancer and the overall efficacy of PDT treatment [15]. Generally, the type of cell death mechanism triggered by PDT is determined by the intracellular localization of the photosensitizer in cancerous cells [43]. Photosensitizers that localize in the mitochondria are inducers of apoptosis; intracellular localization of photosensitizers to cell membrane causes necrosis and delayed induction of apoptosis [47,48]. Studies by Chizenga and colleagues have reported evidence of mitochondrial and lysosomal localization of AlPcSmix photosensitizer, a derivative of AlClPc similar to the as-synthesized AlClPcTS41, in cervical cancer [15]. To evaluate the intracellular localization of AlClPcTS41 and AlClPcTS41-PEG-CuAuNPs, in Caco-2 cells, the organelle-specific markers DAPI (stain nucleus), mitochondrial, and lysosomal trackers were used and the localization of AlClPcTS41 (under selected standard doses of 0.58 µM concentration) was observed in the fluorescence microscopy after uptake of AlClPcTS41-PEG-CuAuNPs. As shown in Figure 10, cells treated with AlClPcTS41-PEG-CuAuNPs displayed a strong yellow fluorescence, as red and green fluorescence overlapped after 4 h of incubation with the stains, suggesting evidence of a co-localization of the PS predominantly in the mitochondria site, as well as localization in the lysosomes. The red fluorescence of the AlClPcTS41 was not observed in the nucleus (stained blue with DAPI). As the primary target of our photosensitizer is the mitochondria and lysosomes, which are closely related to photo induce cell death and apoptosis in PDT-mediated therapy, these results validated the possibility of effective cancer cell death in mitochondria- and lysosome-associated pathways, resulting from PDT with AlClPcTS41-PEG-CuAuNPs, as seen in Figure 10.

PDT uses a combination of photo-activatable photosensitizers in the presence of molecular oxygen to create cytotoxic ROS species [49]. The cellular damage and cell death are the most common cellular responses to PDT-mediated treatment [49]. In several studies, Caco-2 cells underwent morphological alteration such as rounding up in shape, nuclear damage, and detachment of cells after exposure to PDT using second-generation phthalocyanine photosensitizers and their derivatives [4,50]. These findings agree with our study whereby Caco-2 cells displayed significant morphological changes such as cellular shrinkage, cell rounding up, and cell detachment, post-PDT treatment with AlClPcTS41-PEG-CuAuNPs when compared to control groups of Caco-2 cells. Based on these results, we suggest that cytotoxicity was induced by AlClPcTS41-PEG-CuAuNPs + PDT.

To evaluate the ability of AlClPcTS41, PEG-CuAuNPs, and AlClPcTS41-PEG-CuAuNPs to induce in vitro Caco-2 cancer cell death, we utilized laser irradiation at 636 nm and, to further elucidate the cytotoxicity of AlClPcTS41, PEG-CuAuNPs, and AlClPcTS41-PEG-CuAuNPs, LDH assays were then measured. The studies showed that the AlClPcTS41 was non-toxic for Caco-2 when tested in different concentrations, including (0.58 µM) in the absence of laser light-induced activation. At the same time, the dose-dependent killing of Caco-2 (Figure 6) was observed upon laser irradiation of AlClPcTS41 only. The AlClPcTS41-PEG-CuAuNPs in the absence of laser light activation exhibited lower cytotoxicity on colorectal cancer cell lines for concentrations. However, under different conditions, the photodynamic effect of AlClPcTS41-PEG-CuAuNP-mediated treatment was achieved upon laser light treatment, which induced a significant increase in LDH release compared to the control populations or the AlClPcTS41 only. These results indicate that AlClPcTS41-PEG-CuAuNPs is an active photosensitizer, and further investigation for photodynamic antitumor effects is warranted. 

One of the intriguing results of the ATP cell proliferation analysis was the PDT-mediated effects of the novel AlClPcTS41-PEG-CuAuNP. When Caco-2 cells were incubated with AlClPcTS41-PEG-CuAuNP + PDT, cell proliferation was remarkably reduced, compared to the cells-only control group. Considering that the AlClPcTS41 displayed anti-tumor properties, our findings are suggestive that the AlClPcTS41-PEG-CuAuNPs could possibly exhibit enhanced cytotoxic effects and a significant reduction in cell proliferation, thus improving the anti-tumor abilities of PDT.

Studies have reported that the generation of reactive oxygen species (ROS) in the process of PDT treatment plays a critical role in imparting oxidative damage to cell organelles, such as cell membranes, mitochondria, and lysosomes, which can induce cell death in cancerous cells [51]. In this sense, ROS generation analysis, as seen in Figure 15, and assessment of the mode of cell death (necrosis or apoptosis), as seen in Figure 14 after AlClPcTS41, PEG-CuAuNPs, and AlClPcTS41-PEG-CuAuNP-PDT, was performed and read using the VICTOR Nivo^®^ multimode plate reader and a flow cytometer, respectively. Our findings revealed that AlClPcTS41-PEG-CuAuNP-PDT significantly elevated ROS intensity and induced apoptotic (early and late) cell death on Caco-2 cells, as compared to cells only, AlClPcTS41, and PEG-CuAuNPs. Based on these results, it may be suggestive that apoptotic cell death was the likely mechanism by which PDT of AlClPcTS41 induced cell death of Caco-2 when alone and when in conjugates (AlClPcTS41-PEG-CuAuNPs).

## 4. Materials and Methods

### 4.1. Materials 

Phosphate-buffered saline (PBS, 10× concentrate) and formaldehyde (for molecular biology, 36.5–38% in H_2_O) were purchased from (Sigma Aldrich, St. Louis, MI, USA). Human colon cancer cells (CaCo-2 Cellonex Cat SS1402 CCAC-FL; CCAC-C) were purchased from the (American Type Culture Collection, Manassas, VA, USA). Tetrahydrofuran (THF), 1,8-diazabicyclo [5.4.0] undec-7-ene (DBU), aluminum chloride, dimethyl sulfoxide (DMSO), dimethyl formamide (DMF), sodium 2-mercaptoacetate, deuterated water (D_2_O), 4-nitrophthalonitrile, dichloromethane (DCM), methanol, 1-pentanol, *N*-hydroxysuccinimide (NHS), amino-polyethylene glycol-thiol (NH_2_-PEG(2000)-SH), and *N*,*N*′-dicyclohexylcarbodiimide (DCC) were purchased from Sigma–Aldrich. Sodium 2-mercaptoacetate phthalonitrile was synthesized as previously reported [52]. 

### 4.2. Synthesis of AlClPcTS41 and AlClPcTS41-PEG-CuAuNPs

#### 4.2.1. Synthesis of Aluminum (II) Chloride 2(3), 9(10), 16(17), 23(24)-Tetrakis-(sodium 2-Mercaptoacetate) Phthalocyanine

Figure 1. The synthesis of the aluminum (II) chloride 2(3), 9(10), 16(17), 23(24)-tetrakis-(sodium 2-mercaptoacetate) phthalocyanine (AlClPcTS41) was achieved through cyclotetramerization of sodium 2-mercaptoacetate phthalonitrile (500 mg, 2.08 mmol) in a round-bottomed flask containing 3 mL of 1-pentanol and aluminum (III) chloride (50 mg, 0.312 mmol). DBU was added dropwise while stirring at 160 °C for 7 h to afford the AlClPcTS41 complex. Column chromatography using silica as the solid phase and DCM. An amount of 5% water and 10% methanol as the mobile phase were used to isolate the AlClPcTS41 complex from the mixture of unreacted materials. FTIR, NMR, MS, and CHNS elemental analysis were used to confirm the structure of AlClPcTS41.

**AlClPcTS41**. Yield: 69%. Uv–Vis λmax/nm, DMSO. 715 (5.4). 665 (5.1). 365 (4.6). FTIR (KBR), cm^−1^: 3315 (-OH), 1716 (C=O), 1560 (Ar-C=C), 1610 (C=C), 2735 (Ar-S-C), 2950 (Ar-CH). 1H NMR (500 MHz and D2O) δ 8.44 (s, 4H, and -OH), 8.13 (d, 2H, and Ar-H), 7.96 (dd, 1H, and Ar-H), 7.74 (d, 1H, and Ar-H), 7.63 (dd, *J* = 19.3, 11.0 Hz, 8H, and Ar-H), and 3.59 (s, 8H, and CH). MALDI-TOF MS (*m*/*z*). Calc: 1023.28. Found 988.45 [M-Cl(35)]. Calc for C_40_H_20_N_8_Na_4_O_8_S_4_AlCl: C (46.95), H (1.97), N (10.95), and S (12.53). Found C (46.21), H (2.11), N (10.42), and S (13.02).

#### 4.2.2. Synthesis of PEGylated Copper–Gold Bimetallic Nanoparticles 

Figure 2. Synthesis of PEGylated copper–gold bimetallic (alloyed) nanoparticles (PEG-CuAuNPs) were prepared using previously reported methods [53,54] with minor modification. Typically, two aqueous CTAB solutions (10.0 mM), one containing hydrazine (40.0 mM) and the other containing cupric chloride (1.0 mM, pH 10–11) were mixed for 2 h in a round-bottomed flask while stirring at room temperature in an inert environment, with an additional 10 mL of hydrazine added dropwise for 1 h. A solution of chloroauric acid (HAuCl_4_) (50 mL of 0.01 wt%) was added dropwise while stirring at 120 °C for 1 h to load the gold metal to the copper metallic nanoparticle to form an intermetallic (alloyed) nanoparticle [20]. A total of 4.5 mL aliquot of 1 wt% sodium citrate solution was then added to the boiling reaction mixture, and the heating was continued under reflux for 30 min to enable a complete reaction. To enable ligand exchange between sodium citrate and polyethylene glycol (HS-PEG2000-NH_2_), the temperature of the reaction was then lowered to room temperature before adding PEG (10 wt%) dropwise while stirring for 2 h. The newly prepared PEG-CuAuNPs were isolated and washed through centrifugation using ethanol and deionized water.

#### 4.2.3. Conjugation of AlClPcTS41 to PEG-CuAuNPs 

Figure 3. An amide bond linkage between the AlClPcTS41 was achieved through a two-step reaction that involved the activation of the carboxylate salt group of the AlClPcTS41 followed by a reaction with an amino group with the PEG molecules on the surface of copper–gold alloy nanoparticles. Typically, AlClPcTS41 (20 mg, 0.023 mmol), DCC (8.4 mg, 0.040 mmol), and NHS (for 4.61 mg, 0.040 mmol) were added to dry DMF (3 mL) in a vial followed by stirring for 48 h at room temperature to allow activation of the carboxylate salt group on the AlClPcTS41 complex. After this time, PEG-CuAuNPs (20 mg) in dry DMF (2 mL) was added and the mixtures were left stirring for a further 48 h to allow the formation of the amide bond linkage between the NH_2_ groups of PEGylated copper–gold alloy nanoparticles and the carboxylate group of the AlClPcTS41 to form the AlClPcTS41-PEG-CuAuNPs conjugate. The conjugates were isolated and purified from the mixture by repeated centrifuging with DMF, methanol, and ethanol. FTIR was used to confirm the functional groups and the amide bond between the AlClPcTS41 and PEG-CuAuNPs. TEM was used to study the size and morphology of the PEGylated nanoparticles and conjugates.

### 4.3. Cell Culture 

In a 75 cm^2^ tissue culture flask, Caco-2 cells (human colorectal cancer) were grown at 37 °C under a 5% CO_2_ atmosphere and Dulbecco’s Modified Eagle’s medium (DMEM) supplemented with 10% (*v*/*v*) fetal bovine serum (FBS100), penicillin, and 100 μg/mL streptomycin solution, 2.5 μg/mL amphotericin B, and 1 mM sodium pyruvate was used as culture medium. On every alternate day, the culture medium was changed to a fresh medium. Confluent cell monolayers were detached by trypsin and cell suspension seeded at 6 × 10^5^ cells in 3.5 cm^2^ diameter culture plates and maintained for in vitro cellular experimental purposes.

### 4.4. Photodynamic Therapy 

The Caco-2 colorectal cells were grown in a culture medium and upon reaching confluency, PBS was used to wash cells that were detached from the culture flask (0.25% trypsin). Cells were then seeded into culture plates. Cells were divided into various groups: untreated groups, dark (no irradiation/control group), and cells incubated with different AlClPcTS41, PEG-CuAuNPs and AlClPcTS41-PEG-CuAuNPs under laser light irradiation (636 nm, fluency 10 J/cm^2^). After treatment, the cells were incubated for 24 h for further biochemical experiments.

#### PDT Dose Response

We first evaluated cellular morphology changes by adopting methodology from (Section 4.6.1). The viability and cytotoxicity of the AlClPcTS41 in Caco-2 cells were then determined using the Trypan Blue and LDH assay, respectively, post-treatment, to determine the AlClPcTS41 IC50 inhibitory concentration with varying concentrations of AlClPcTS41 (0.125, 0.25, 0.5 and 0.75 µM). Cell culture was performed as previously mentioned (Section 4.3) and a laser was admitted, as per (Section 4.4). The cell viability was evaluated by incubating the cells with trypan blue. Briefly, trypan blue dye, a viability exclusion assay, allows for the enumeration of the viable cells (percentage) in the cell suspension. Equal volumes (20 µL) of 0.4% (*w*/*v*) trypan blue dye (Invitrogen, Trypan Blue Stain Thermo Fisher-T10282) were added to (20 µL) cell suspensions and quantified using an automated cell counter (10 µL). 

### 4.5. AlClPcTS41 and AlClPcTS41-PEG-CuAuNPs Subcellular Localization Studies

Images were acquired with fluorescence microscopy to determine the AlClPcTS41 and AlClPcTS41-PEG-CuAuNPs cellular studies and localization experiments in Caco-2 cells. The cells were cultured (6 × 10^5^ cells) on sterile coverslips in 3.5 cm^2^ diameter culture plates at 37 °C overnight under incubation for cell attachment, followed by the addition of the AlClPcTS41, PEG-CuAuNPs, and AlClPcTS41-PEG-CuAuNPs incubated for 4 h in fresh media. Caco-2 cells were then washed 2X with PBS, fixed with 4% formaldehyde, and stained with 100 nM FITC-Mitotracker and 65 nM lysosomal (Lyso-Tracker). The cell nuclei were counterstained with 4′,6-diamidino-2-phenylindole (2.5% DAPI) in the dark. The slides were followed by observation with (Alexa Fluor 594, DAPI, and FITC filters).

### 4.6. AlClPcTS41 and AlClPcTS41-PEG-CuAuNPs-Mediated PDT

#### 4.6.1. Morphology Assessment

Cellular morphological changes resulting from PDT treatment were observed 24 h post-treatment using an inverted microscope (Wirsam, Olympus CKX41), fitted with a digital SC30 Olympus camera at 200× magnification.

#### 4.6.2. Cytotoxicity

Cellular cytotoxicity was studied using the CytoTox96^®^ Non-Radioactive Cytotoxicity Assay (Promega G1780- Promega Corp, Madison, WI, USA) based on the manufacturer’s instructions. VICTOR Nivo^®^ multimode plate reader (PerkinElmer, HH35940080 EN) was used to monitor the LDH release at 490 nm during membrane damage as compared to the positive control at 100% cytotoxicity.

#### 4.6.3. Proliferation 

The CellTiter-Glo^®^ Luminescent cell viability assay (Promega, G7570) was utilized to determine cell proliferation of Caco-2 cells (the number of metabolically viable cells) after 24 h of incubation and treatment. The luminescent signal of the thermostable luciferase enzyme was used to determine the metabolic activity of cells, indicative of the amount of ATP within cells. Typically, 50 µL of cell suspension and 50 µL of CellTiter-Glo reagent were added to 96 multi-well plates (opaque-walled). After 10 min of incubation in the dark, ATP luminescence was then measured with the VICTOR Nivo^®^ multimode plate reader (PerkinElmer, HH35940080 EN).

#### 4.6.4. Cell Death Assay

Quantitative analysis of apoptosis effects induced by PDT treatment-mediated AlClPcTS41-PEG-CuAuNPs was assessed by flow cytometry using the Annexin V-FITC/PI Apoptosis Detection Kit. In brief, culture plates were used to seed Caco-2 cells and incubated overnight, followed by media change, and replaced with free AlClPcTS41, PEG-CuAuNPs, and AlClPcTS41-PEG-CuAuNPs for another 4 h. PBS was then used to wash cells before laser irradiation (636 nm, 10 J/cm^2^). Control studies were performed without irradiation and kept in the dark. Cells were washed with PBS After 24 h incubation, detached, and stained with 5 μL FITC Annexin-V and 5 μL PI reagents, sequentially. Lastly, a 400 μL binding buffer was used to disperse the cells and was then incubated for 15 min at room temperature in the dark, followed by analysis using a flow cytometer.

#### 4.6.5. Intracellular Assessment of Reactive Oxygen Species

The ROS-Glo H_2_O_2_ assay was used to assess ROS generated in Cac0-2 cells after treatment with AlClPcTS41/AlClPcTS41-PEG-CuAuNPs in the presence/absence of 636 nm light irradiation. Briefly, the Caco-2 cells were seeded in a 96-well plate with 2 × 10^3^ cells/well in 100 µL DMEM. After 24 h of incubation, the cells were treated with fresh media containing AlClPcTS41/AlClPcTS41-PEG-CuAuNPs under an equivalent concentration of AlClPcTS41 and were further incubated for 4 h. The cells were then irradiated with a 636 nm laser. After incubation for 18 h, the ROS assay was added in a final concentration of 25 µM, in accordance with the protocol of the manufacturer’s manual and, 6 h later, an ROS-Glo detection reagent (100 µL) was added to each well for a further 20 min. Relative luminescence units were recorded using the VICTOR Nivo^®^ multimode plate reader (PerkinElmer, HH35940080 EN).

### 4.7. Statistical Analysis 

All the biochemical assay data presented in the study were carried out from at least three independent experiments. We used the Sigma Plot version 14.0. software to statistically analyze the data. Comparisons were made using the one-way analysis of variances (ANOVA) for normally distributed data. Asterisks indicate significant differences with respect to control and experimental groups, * *p* < 0.05, ** *p* < 0.01, and *** *p* < 0.001 (one-way ANOVA).

## 5. Conclusions

In conclusion, we reported for the first time the design, synthesis, and characterization of AlClPcTS41 and further conjugated it to PEG-CuAuNPs. PDT-mediated anti-tumor effects of the newly synthesized AlClPcTS41 when alone and when conjugated to PEGylated nanoparticles were also reported for the first time within this study. The nanoparticles and the conjugates were characterized successfully using UV–VIS, FTIR, XRD, TEM, and DLS, and were further applied for photodynamic therapy of colon cancer (Caco-2). The novel AlClPcTS41 and the conjugate (AlClPcTS41-PEG-CuAuNPs) demonstrated significant ROS generation abilities that were used for the efficacy of PDT on Caco-2. The effectiveness of using AlClPcTS41 and AlClPcTS41-PEG-CuAuNPs-mediated PDT was shown in inverted microscopy images, where a significant alteration in the cell morphology, as well as a greater population of Caco2 cells treated with the composite dying, were the characteristics observed. Our choice of nanoparticle system, pegylated copper–gold bimetallic (alloy) nanoparticles (PEG-CuAuNPs) for delivering AlClPcTS41, increased the uptake of the PS into Caco-2 cells, resulted in a significant decrease in cell viability with AlClPcTS41-PEG-CuAuNPs-PDT-mediated treatment when compared to AlClPcTS41-PDT or without irradiated Caco2 cells. The dark toxicity experiments demonstrated that the designed AlClPcTS41-PEG-CuAuNPs showed negligible cytotoxicity; it was biocompatible. Finally, flow cytometry analysis of treated Caco-2 cells confirmed the obtained results. Increased cell death populations were observed. The lack of dark cytotoxicity of AlClPcTS41 offered promising possibilities in application with PEG-CuAuNPs nanoparticles for effective photodynamic therapy in Caco-2 cells.

## Data Availability

Not applicable.

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
