# Peer review of "Photodynamic Therapy of Aluminum Phthalocyanine Tetra Sodium 2-Mercaptoacetate Linked to PEGylated Copper–Gold Bimetallic Nanoparticles on Colon Cancer Cells"

_ijms, 2023, doi:10.3390/ijms24031902_

Round 1

Reviewer 1 Report

Authors must describe the biological windows (or at least the first) in the sentence "AlClPc present a very strong absorption within the 650−800 nm, allowing for efficient 64 treatment of cancerous tissues with negligible phototoxicity"

Authors describe "The thiol group (sulphur) of the MPcs can 73 further improve their water solubility and prevent them from aggregating, without sig-74 nificantly hampering their photophysical properties" but the sentence should be better explained. Thiol group are used to link compounds to Au/Cu NPs, and therefore, these groups could not help in water stability.

The sentence "Gold 80 nanoparticles (AuNPs) have already been established in PDT-induced colorectal cancer 81 treatments because of their distinctive miniature dimensions and large surface-area-to-82 volume ratio" must be corrected. These skills is not only for gold NPs.

The paragraph "These nanoparticles (NPs) in different forms may passively localize in 83 cancerous cells due to the tumour characteristics such as leaky vasculature and poor lym-84 phatic drainage, a phenomenon generally known as enhanced permeability and retention 85 effect (EPR)" must be corrected, EPR effect presents some limitations. DOI: 10.1039/d1bm01398j and DOI: 10.1016/j.jconrel.2014.03.057

The paragraph "Furthermore, to address the serious biological barriers such as mac-86 rophages, often encountered by NPs, polyethylene glycol (PEG) used as surface modifi-87 cations polymer can help improve the solubility of nanoparticles, reduce their nonspecific 88 interactions with the biological barriers, improve accumulation target disease, reduce cell 89 binding , uptake as well as interfere with aggregation, thus enhancing surface hydro-90 philicity" needs more references. DOI: 10.1039/c7nr01110e

Based on references 23-24 and the TEM provided by authors, they should change core-shell structure to metal alloys. Neither reference 24 or authors demonstrated a real core-shell structure.

In addition to FTIR, the covalent link between PEG-CuAuNP and AlClPcTS41 must be studied by NMR. Since CuAuNPs are not magnetic, it will help to confirm the successful link. 

Line 303, "Figure e." Could be a mistake?

AlClPcTS41-PEG-311 CuAuNPs looks aggregated. A DLS study must be conducted in order to check their colloidal stability

In the UV-Vis, the spectrum of AlClPcTS41-PEG-CuAuNPs looks too diluted (based on signal to noise ratio) and later multiplied. Could authors confirm this fact. 

Could authors explain why in Figure 7 , NPs looks close to the nuclei? Moreover, a ICP analysis about the uptake of NPs must be conducted.

There is lack of Trypan Blue analysis with AlClPcTS41-PEG-CuAuNPs. Moreover, it will be better if authors check the organic amount of NPs (TGA) and compare AlClPcTS41-PEG- CuAuNPs with AlClPcTS41 and laser. CuAu NPs present a moderate absorption in the NIR

The reviewer believe that a merge of all "cells results" in one single Figure with all techniques will be better to understand the results. In separated figures is not easy to check/compare the results. 

AlClPcTS41 mainly increase the intracellular ROS levels, but authors did not check anything about ROS. Please do it. 

Author Response

REVIEWER 1

 1.Authors must describe the biological windows (or at least the first) in the sentence "AlClPc present a very strong absorption within the 650−800 nm, allowing for efficient treatment of cancerous tissues with negligible phototoxicity"

AUTHOR RESPONSE: Thank you for pointing this out. Accordingly, we have revised and modified the paragraph. It has been changed to “AlClPc present strong absorption of visible light at the wavelength between 650−800 nm where tissue penetration is maximized, which is crucial for PDT, since it can allow for efficient treatment of cancerous tissues with negligible phototoxicity”. (L64-L66).

  1. Authors describe "The thiol group (sulphur) of the MPcs can 73 further improve their water solubility and prevent them from aggregating, without sig-74 nificantly hampering their photophysical properties" but the sentence should be better explained. Thiol group are used to link compounds to Au/Cu NPs, and therefore, these groups could not help in water stability.

AUTHOR RESPONSE: Thank you, although this group is involved in bonding, its presence in a tetra form on the Pc ring and on the PEG molecules still facilitates water solubility due to increased polarity.

  1. The sentence "Gold 80 nanoparticles (AuNPs) have already been established in PDT-induced colorectal cancer 81 treatments because of their distinctive miniature dimensions and large surface-area-to-82 volume ratio" must be corrected. These skills is not only for gold NPs.

AUTHOR RESPONSE: Thank you for pointing this out and we agree in this. The paragraph has been revised and now reads” Among the metallic nanoparticles, one of the increasingly favourable PS carriers within PDT-induced colorectal cancer treatment, are gold nanoparticles (AuNPs), principally because they possess unique properties such as distinctive miniature dimensions and large surface-area-to-volume ratio.” (L82-85).

  1. The paragraph "These nanoparticles (NPs) in different forms may passively localize in 83 cancerous cells due to the tumour characteristics such as leaky vasculature and poor lym-84 phatic drainage, a phenomenon generally known as enhanced permeability and retention 85 effect (EPR)" must be corrected, EPR effect presents some limitations. DOI: 10.1039/d1bm01398j and DOI: 10.1016/j.jconrel.2014.03.057

AUTHOR RESPONSE: Thank you for pointing this out.  L85-L91 have been revised and now reads “These nanoparticles (NPs) in different forms have been considered to passively localize in cancerous cells owing to certain tumour characteristics such as leaky vasculature and poor lymphatic drainage, a phenomenon generally known as enhanced permeability and retention effect. Studies by Caro et al (2021) have also reported that the EPR effect is also highly dependent on the type of tumour and tumour site, as some factors to consider for efficient tumour targeting strategy.

  1. The paragraph "Furthermore, to address the serious biological barriers such as mac-86 rophages, often encountered by NPs, polyethylene glycol (PEG) used as surface modifi-87 cations polymer can help improve the solubility of nanoparticles, reduce their nonspecific 88 interactions with the biological barriers, improve accumulation target disease, reduce cell 89 binding, uptake as well as interfere with aggregation, thus enhancing surface hydro-90 philicity" needs more references. DOI: 10.1039/c7nr01110e

AUTHOR RESPONSE: Noted with thanks. We are grateful to the reviewers for their insightful comments. Additional reference DOI: 10.1039/c7nr01110e, has been added.

  1. Based on references 23-24 and the TEM provided by authors, they should change core-shell structure to metal alloys. Neither reference 24 or authors demonstrated a real core-shell structure.

AUTHOR RESPONSE: Thank you, new TEM images have been acquired and now have clear demonstration of the core and shell of the NPs, as well as after conjugation with AlPc, this causes aggregation and have been explained with reference in text.

  1. In addition to FTIR, the covalent link between PEG-CuAuNP and AlClPcTS41 must be studied by NMR. Since CuAuNPs are not magnetic, it will help to confirm the successful link. 

AUTHOR RESPONSE: Thank you, the current techniques does show the amide bond between the nanoparticle and the phthalocyanine. The best technique, that’s more sensitive would be XPS, which we would consider in our future studies.

 8.Line 303, "Figure e." Could be a mistake?

AUTHOR RESPONSE: Noted with thanks. Figure “e” has been modified and corrected, Figure 2A. (L305).

  1. AlClPcTS41-PEG-311 CuAuNPs looks aggregated. A DLS study must be conducted in order to check their colloidal stability

AUTHOR RESPONSE: Thank you, we have added DLS and the PDI values to demonstrate the colloidal stability and hydrodynamic sizes, summarized in table 1 and on micrographs.  

  • In the UV-Vis, the spectrum of AlClPcTS41-PEG-CuAuNPs looks too diluted (based on signal to noise ratio) and later multiplied. Could authors confirm this fact.

AUTHOR RESPONSE: Thank you, we have now replaced the spectrum of AlClPcTS41-PEG-CuAuNPs with a better concentrated spectrum

  • Could authors explain why in Figure 7, NPs looks close to the nuclei? Moreover, a ICP analysis about the uptake of NPs must be conducted.

AUTHOR RESPONSE: Thank you, uptake studies were repeated and are correctly demonstrated on figure 10.

  • There is lack of Trypan Blue analysis with AlClPcTS41-PEG-CuAuNPs. Moreover, it will be better if authors check the organic amount of NPs (TGA) and compare AlClPcTS41-PEG- CuAuNPs with AlClPcTS41 and laser. CuAu NPs present a moderate absorption in the NIR

AUTHOR RESPONSE: Noted with thanks. Accordingly, we have conducted Trypan Blue analysis with AlClPcTS41-PEG-CuAuNPs.

  • The reviewer believe that a merge of all "cells results" in one single Figure with all techniques will be better to understand the results. In separated figures is not easy to check/compare the results.

AUTHOR RESPONSE: Thank you for this suggestion. Although we agree that it would have been interesting to explore this aspect, however, using a merge of all “cells results” in one single Figure would not effectively represent the results. For example, the trypan blue assays were merely used as dose response assays to determine the ICD50 for the AlClPcTS41 PS, and, within the final outcomes of the AlClPcTS41-PEG-CuAuNPs- experiment, the assays confirm the significant contribution of PS drug carriers.

  • AlClPcTS41 mainly increase the intracellular ROS levels, but authors did not check anything about ROS. Please do it.

AUTHOR RESPONSE: Noted with thanks. We have added the context suggested by reviewer, the intracellular ROS level analysis was conducted. ROS methodology and Result section were added. Furthermore, L 579- L584, was revised and now read” In this sense, ROS generation analysis and assessment of mode of cell death (necrosis or apoptosis) after AlClPcTS41, PEG-CuAuNPs and AlClPcTS41-PEG-CuAuNP –PDT was performed and read using the VICTOR Nivo® multimode plate reader and a flow cytometer, respectively. Our findings revealed that AlClPcTS41-PEG-CuAuNP-PDT significantly elevated ROS intensity and induced apoptotic (early and late) cell death on Caco-2 cells, as compared to cells only, AlClPcTS41 and PEG-CuAuNPs”.

Reviewer 2 Report

Abrahamse et al. has studied photodynamic therapy on colon cancer cells using AICIPcTS41-PEG-CuAuNPs. The authors have investigated PDT dose response in a nice detailed way. However, I have some concerns regarding the characterization of nanoparticles. TEM is the most important characterization for nanoparticles. The TEM images shown in the manuscript do not suggest a good quality nanoparticle. In the case of PEG-Cu-Au NPs, at least a few good quality nanoparticles are visible, however size distribution is really poor. The TEM image shown for AICIPcTS41-PEG-CuAuNPs do not suggest retention of a nanoparticle structure, it clearly shows aggregation and retention of a lot of solvent. Authors should prepare TEM sample very carefully, a good time sonication before sampling and proper drying of solvent after sampling will help. Authors must redo the TEM experiment for both of the NPs. HRTEM and SAED pattern should be included with TEM, it will be more helpful in detailed characterization of the NPs. In addition, authors should also perform DLS study for the NPs for a better picture of size distribution.

The UV-Vis study is very informative. I have a small concern on the UV-Vis figure of AICIPcTS41-PEG-CuAuNPs, it looks very noisy. Authors should repeat the UV-Vis for AICIPcTS41-PEG-CuAuNPs with slightly higher concentration and a slow scan rate. In case the noise is for aggregation, sonication before performing the experiment will be helpful.

Additional characterization of the nanoparticles e.g. XRD, XPS, EDS will be highly appreciated.

Author Response

REVIEWER 2

  1. Abrahamse et al. has studied photodynamic therapy on colon cancer cells using AICIPcTS41-PEG-CuAuNPs. The authors have investigated PDT dose response in a nice detailed way. However, I have some concerns regarding the characterization of nanoparticles. TEM is the most important characterization for nanoparticles. The TEM images shown in the manuscript do not suggest a good quality nanoparticle. In the case of PEG-Cu-Au NPs, at least a few good quality nanoparticles are visible, however size distribution is really poor. The TEM image shown for AICIPcTS41-PEG-CuAuNPs do not suggest retention of a nanoparticle structure, it clearly shows aggregation and retention of a lot of solvent. Authors should prepare TEM sample very carefully, a good time sonication before sampling and proper drying of solvent after sampling will help. Authors must redo the TEM experiment for both of the NPs. HRTEM and SAED pattern should be included with TEM, it will be more helpful in detailed characterization of the NPs. In addition, authors should also perform DLS study for the NPs for a better picture of size distribution

AUTHOR RESPONSE: Thank you, a clear image of TEM has been acquired and added to replace the previous. We have also added the XRD and DLS of the nanoparticle and the conjugate.

  1. 2. The UV-Vis study is very informative. I have a small concern on the UV-Vis figure of AICIPcTS41-PEG-CuAuNPs, it looks very noisy. Authors should repeat the UV-Vis for AICIPcTS41-PEG-CuAuNPs with slightly higher concentration and a slow scan rate. In case the noise is for aggregation, sonication before performing the experiment will be helpful.

AUTHOR RESPONSE: Thank you, a new spectrum of the conjugate has been added to replace the previous.

  1. Additional characterization of the nanoparticles e.g. XRD, XPS, EDS will be highly appreciated.

AUTHOR RESPONSE: Thank you, we have corrected the TEM image and also added DLS and XRD, we believe this should be sufficient to demonstrate the link, crystallinity, size and morphology of the nanoparticle and the conjugates.

Round 2

Reviewer 1 Report

The reviewer believe that authors have solved all issues in the work.

Just really minor things:

1. Quality of figures. I guess that the final version will have High Resolution figures, and this is only for reviewers. I fnot, please improve them.

2. Figure 4. Please, it would better if X axis have the same tick label (For example both every 20 nm or both every 10 nm)

Author Response

The reviewer believe that authors have solved all issues in the work.

AUTHOR RESPONSE. Thank you for the time invested in reviewing the MN, in addition to the comments addressed below, we have also rigorously made language changes and spell checks across the MN to improve its quality.

Just really minor things:

  1. Quality of figures. I guess that the final version will have High Resolution figures, and this is only for reviewers. I fnot, please improve them.

AUTHOR RESPONSE. Thank you, we have provided original figures with high resolution to be used in the final MN.

  1. Figure 4. Please, it would better if X axis have the same tick label (For example both every 20 nm or both every 10 nm).

AUTHOR RESPONSE. Thank you, we have made the changes as suggested and replaced the figure with an edited version.

Reviewer 2 Report

The authors have addressed to most of the reviewer concerns successfully. However, I still do not believe the structure of nanocomposite to be a core-shell structure. The new TEM image shows black core surrounded by white shell. This is not actually an ideal example of core-shell structure. The core and shell do not have much contrast in color, usually cores are black surrounded by a grey shell. The white shell as shown by authors is actually shell of solvents. Moreover, the XRD shows peak of Cu and Au, the peak height ratio is almost equal. Therefore, authors should change their title and modify their manuscript as alloyed metal nanoparticles instead of core-shell metal nanoparticles.

Author Response

Reviewer 2.

The authors have addressed to most of the reviewer concerns successfully. However, I still do not believe the structure of nanocomposite to be a core-shell structure. The new TEM image shows black core surrounded by white shell. This is not actually an ideal example of core-shell structure. The core and shell do not have much contrast in color, usually cores are black surrounded by a grey shell. The white shell as shown by authors is actually shell of solvents. Moreover, the XRD shows peak of Cu and Au, the peak height ratio is almost equal. Therefore, authors should change their title and modify their manuscript as alloyed metal nanoparticles instead of core-shell metal nanoparticles.

AUTHOR RESPONSE. Thank you, we have made the changes on the title and across the manuscript to highlight the structure of nanocomposite as bimetallic (intermetallic alloy).